# Overfitting: An Unexpected Asset in AI-Generated Image Detection

## Abstract

AI-generated images have become highly realistic, raising concerns about potential misuse for malicious purposes. In this work, we propose a novel approach, DetGO, to detect generated images by overfitting the distribution of natural images. Our critical insight is that a model overfitting to one distribution (natural images) will fail to generalize to another (AI-generated images). Inspired by the sharpness-aware minimization, where the objective function is designed in a min-max scheme to find flattening minima for better generalization, DetGO instead seeks to overfit the natural image distribution in a max-min manner. This requires finding a solution with a minimal loss near the current solution and then maximizing the loss at this solution, leading to sharp minima. To address the divergence issue caused by the outer maximization, we introduce an anchor model that fits the natural image distribution. In particular, we learn an overfitting model that produces the same outputs as the anchor model while exhibiting abrupt loss behavior for small perturbations. Consequently, we can effectively determine whether an input image is AI-generated by calculating the output differences between these two models. Extensive experiments across multiple benchmarks demonstrate the effectiveness of our proposed method.

## 1 Introduction

The rapid advancement of generative models (Ho et al., 2020; Song et al., 2021; Gu et al., 2022; Liu et al., 2022; Rombach et al., 2022; Midjourney, 2022) has revolutionized the field of image synthesis, allowing the creation of highly realistic images that are increasingly difficult to distinguish from those captured in the real world. This unprecedented ability to generate photorealistic images has sparked significant interest across various domains, ranging from creative industries to scientific research. However, alongside these exciting possibilities comes a growing concern over the potential for misuse, particularly in the context of misinformation (Qi et al., 2019), fraud (Uyyala & Yadav, 2023), and malicious activities like deepfake generation (Fanelli, 2009). As these synthetic images become more sophisticated, the line between real and generated content blurs, raising critical questions about authenticity and trust in digital media. This growing threat has underscored the urgent need for effective techniques to differentiate between authentic and AI-generated images reliably.

Traditional methods (Frank et al., 2020; Dzanic et al., 2020; Sinitsa & Fried, 2023; Qian et al., 2020) for detecting AI-generated images have focused mainly on identifying visual artifacts or inconsistencies that are inadvertently introduced during the image generation process. These approaches, which typically rely on training classifiers to recognize such anomalies, require large datasets containing both real and generated images. However, as generative models continue to evolve, these artifacts become increasingly subtle or even nonexistent (Corvi et al., 2023), making it progressively more challenging to identify generated images using conventional techniques. Consequently, there is a pressing need for new detection strategies that are robust to the advances in generative models and do not depend on the existence of easily recognizable artifacts in the generated content.

In this paper, we propose DetGO, a novel detection method that addresses the limitations of traditional approaches by fundamentally shifting the focus from *detecting generation-specific artifacts* to *overfitting the distribution of natural images*. DetGO operates on the critical insight that a model trained to overfit a single distribution—in this case, natural images—will inherently struggle to generalize to another distribution, such as AI-generated images. By focusing on this distributional

mismatch, DetGO is able to detect generated images without requiring access to AI-generated images during training. This is a significant departure from traditional methods that depend on both natural and generated images to build their classifiers. Instead, DetGO capitalizes on the inherent differences between the distributions of real and generated images, providing a more robust and scalable solution as generative models become increasingly sophisticated.

Technically, we draw inspiration from Sharpness-Aware Minimization (SAM) (Foret et al., 2021). Both theoretical and empirical evidence suggests that smoother geometries of the loss landscape, particularly the flatness of minima, often lead to improved generalization performance (Keskar et al., 2017; Dziugaite & Roy, 2017; Jiang et al., 2020). In particular, SAM identifies flatter regions through an initial maximization followed by minimization of the loss, thereby enhancing the model's generalization capability. In contrast, we take the opposite approach by actively seeking sharp minima when it trains models over natural images. This sharpness makes a model fit the natural image distribution tightly, limiting the model's ability to generalize to a different distribution, i.e., generated image distribution. To make the loss landscape sharp, we introduce a novel framework with two models, i.e., an anchor model and an overfitting model. The anchor model is designed to be a non-parametric image encoding function, while the overfitting model is trained to overfit the natural image distribution. In particular, the overfitting model is adjusted to produce outputs that closely match those of the anchor model. However, these two models exhibit drastically difference in loss values under slight perturbations. This divergence allows us to effectively identify AI-generated images, as the generated images are unable to follow the tight distribution that the overfitting model has been trained to capture.

The novel contribution of DetGO lies in its ability to exploit the inherent distribution discrepancy between natural and generated images, providing a detection framework that does not rely on the presence of generation-specific artifacts. Moreover, this approach avoids the need for a large dataset of generated images, which can be challenging to obtain and may not cover the wide range of generative models that continue to emerge. Thus, DetGO offers a scalable and flexible solution that can adapt to new types of generative models without the need for retraining on new generated images. To verify the effectiveness of the proposed DetGO, we present comprehensive experiments across multiple benchmarks. Our experimental results demonstrate that DetGO not only surpasses traditional detection methods but also remains effective with advancements in generative models.

Our main contributions can be summarized as follows:

- We provide a new approach to detect AI-generated images by exploiting the nature of overfitting to natural image distribution. This gets rid of the identification of differences between AI-generated and natural images.

- We propose a novel dual-model framework termed DetGO to exploit the nature of overfitting to natural image distribution for AI-generated image detection. DetGO trains a model to overfit natural image by a max-min scheme, i.e., making models sensitive to slight perturbations, inspired by sharpness-aware minimization.

- Comprehensive experiments on benchmarks demonstrate the effectiveness of the proposed method. Moreover, DetGO exhibits strong robustness to changes in generative models, as its training process eliminates the need for AI-generated image.

## 2 RELATED WORKS

We will begin by reviewing the related achievements of prior research in the detection of generated images. Following this, we will introduce several fundamental concepts underpinning our overfitting principle, which forms the basis of our proposed approach.

**AI-generated images detection.**  With the rapid development of generative models like GAN (Goodfellow et al., 2020) and diffusion (Ho et al., 2020) frameworks, the ability to create realistic synthetic images has surged, necessitating effective detection algorithms. Recent learning-based approaches include CNNspot (Wang et al., 2020), which showed that a simple classifier trained on ProGAN-generated (Karras et al., 2018) images can generalize to unseen GAN outputs with augmentation techniques. DIRE (Wang et al., 2023a) found that diffusion models better reconstruct diffusion-generated images than real ones, training a binary classifier based on reconstruction

errors. Ojha (Ojha et al., 2023) noted that traditional deep learning methods struggle with new generative models, but detection in the CLIP (Radford et al., 2021) feature space can generalize well. NPR (Tan et al., 2023a) utilized the upsampling characteristics of generative models to train a classifier on pixel relationships. Meanwhile, training-free methods like AEROBLADE (Ricker et al., 2024) demonstrated that autoencoders can more accurately reconstruct generated images than real ones. In contrast, DetGO requires no prior knowledge of generative models and is trained solely on real images, achieving strong generalization across benchmarks.

**Overfitting.** Overfitting has traditionally been viewed negatively in classical statistical learning theory, where models with increasing complexity, tend to perform poorly on unseen data. Traditional methods such as regularization techniques (Krogh & Hertz, 1991)), have been widely utilized to combat overfitting by penalizing complex models. Early stopping (Morgan & Bourlard, 1989), another classical technique aimed at halting training before the model starts to overfit, has received less attention in deep learning. The interplay between regularization methods and the generalization capabilities of deep networks has been explored in various studies. For instance, recent work highlights the inadequacy of the classical bias-variance trade-off in explaining the generalization performance of overparameterized models (Zhang et al., 2017), particularly in light of the phenomenon known as "double descent" (Belkin et al., 2018; Nakkiran et al., 2020). This suggests that deeper networks can continue to improve in performance even after achieving perfect training accuracy, a counterintuitive result that challenges traditional views on model complexity. Empirical techniques specifically designed to reduce overfitting in deep learning have also gained prominence. Dropout (Srivastava et al., 2014), a stochastic regularization method that randomly removes units during training, aims to mitigate co-adaptation among neurons, thereby enhancing model robustness. Data augmentation techniques, such as Cutout (Devries & Taylor, 2017) and mixup (Zhang et al., 2018), have been shown to effectively improve generalization by artificially increasing the diversity of the training set. These approaches encourage the model to learn more invariant representations and reduce sensitivity to specific training samples. Studies indicate that these methods specifically designed to combat overfitting are generally less effective in practice than employing early stopping (Rice et al., 2020). In this work, we consider the overfitting to specific distributions as an asset in the context of AI-generated image detection.

## 3 METHOD

### 3.1 MOTIVATION

As discussed in Sharpness-Aware Minimization (SAM), a smoother loss landscape tends to enhance generalization performance. To improve generalization, we aim to find parameter values where the entire neighborhood exhibits both low training loss and low curvature. Specifically, this can be achieved by optimizing the following loss (Foret et al., 2021):

$$\min_{\boldsymbol{w}} L_{\mathcal{S}}^{SAM}(\boldsymbol{w}) + \lambda ||\boldsymbol{w}||_2^2 \ \text{ where } \ L_{\mathcal{S}}^{SAM}(\boldsymbol{w}) \triangleq \max_{||\boldsymbol{\epsilon}||_2 \leq \rho} L_S(\boldsymbol{w} + \boldsymbol{\epsilon}), \tag{1}$$

where $\rho \geq 0$ is a hyperparameter, $\mathcal{S}$ is a training set and $\boldsymbol{w}$ is parameter value of loss $L_{\mathcal{S}}^{SAM}$.

On the contrary, our model seeks to achieve the worst generalization performance by overfitting to the real image distribution, thereby preventing generalization to the generated image distribution. Disregarding the regularization term, we achieve this by reversing the SAM objective:

$$\max_{\boldsymbol{\theta}} L_{\boldsymbol{\theta}}'(\boldsymbol{x}) \ \text{ where } \ L_{\boldsymbol{\theta}}'(\boldsymbol{x}) \triangleq \min_{0 < ||\boldsymbol{\epsilon}||_2 \leq \rho} L_{\boldsymbol{\theta}}(\boldsymbol{x} + \boldsymbol{\epsilon}), \tag{2}$$

where $L_{\boldsymbol{\theta}}(\boldsymbol{x}) : \mathbb{R}^d \rightarrow \mathbb{R}$ represent the loss of the model parameterized by $\boldsymbol{\theta}$ at a data point $\boldsymbol{x} \in \mathcal{X} \subset \mathbb{R}^d$, $d$ denotes the dimension of images.

### 3.2 MINIMIZATION

In order to maximize $L_{\boldsymbol{\theta}}'(\boldsymbol{x})$, we need to take the derivative with respect to its independent variable $\boldsymbol{x}$ to obtain the maximum sharpness. However, directly differentiating $L_{\boldsymbol{\theta}}'(\boldsymbol{x})$ is challenging, so we approach the problem by starting with $L_{\boldsymbol{\theta}}(\boldsymbol{x})$. Since $\boldsymbol{\epsilon}$ is close to $\mathbf{0}$, we perform a first-order Taylor expansion of $L_{\boldsymbol{\theta}}(\boldsymbol{x} + \boldsymbol{\epsilon})$ at the point $\boldsymbol{x}$:

$$L_{\boldsymbol{\theta}}(\boldsymbol{x} + \boldsymbol{\epsilon}) \approx L_{\boldsymbol{\theta}}(\boldsymbol{x}) + \boldsymbol{\epsilon}^T \nabla_{\boldsymbol{x}} L_{\boldsymbol{\theta}}(\boldsymbol{x}). \tag{3}$$

There exists an $\boldsymbol{\epsilon}$ such that $0 < ||\boldsymbol{\epsilon}||_2 \leq \rho$ that minimizes $L_{\boldsymbol{\theta}}(\boldsymbol{x} + \boldsymbol{\epsilon})$, and $\boldsymbol{\epsilon}$ should satisfy the following condition:

$$\hat{\epsilon}(\boldsymbol{\theta}, \boldsymbol{x}) = \underset{0 < ||\boldsymbol{\epsilon}||_2 \leq \rho}{\arg \min} L_{\boldsymbol{\theta}}(\boldsymbol{x} + \boldsymbol{\epsilon}) \approx \underset{0 < ||\boldsymbol{\epsilon}||_2 \leq \rho}{\arg \min} \boldsymbol{\epsilon}^T \nabla_{\boldsymbol{x}} L_{\boldsymbol{\theta}}(\boldsymbol{x}). \tag{4}$$

Substituting $\hat{\epsilon}(\boldsymbol{\theta}, \boldsymbol{x})$ into Equation 2, we get:

$$L'_{\boldsymbol{\theta}}(\boldsymbol{x}) = L_{\boldsymbol{\theta}}(\boldsymbol{x} + \hat{\epsilon}(\boldsymbol{\theta}, \boldsymbol{x})). \tag{5}$$

### 3.3 MAXIMIZATION

Because our model only accepts inputs of real images during training, $\boldsymbol{x}$ represents a real image in this context. When the model receives a real image input $\boldsymbol{x}$, the loss is computed as $L_{\boldsymbol{\theta}}(\boldsymbol{x})$, and from this calculate $L'_{\boldsymbol{\theta}}(\boldsymbol{x})$. We aim to fit $L_{\boldsymbol{\theta}}(\boldsymbol{x})$ to the point of maximum sharpness at $\boldsymbol{x}$, ensuring minimizing $L_{\boldsymbol{\theta}}(\boldsymbol{x})$ while maximizing $L'_{\boldsymbol{\theta}}(\boldsymbol{x})$, i.e., $L_{\boldsymbol{\theta}}(\boldsymbol{x} + \hat{\epsilon}(\boldsymbol{\theta}, \boldsymbol{x}))$. Clearly, we now need to optimize $\boldsymbol{\theta}$ to simultaneously achieve both objectives. Our insight leads to the construction of a new loss function $\mathcal{L}_{\boldsymbol{\theta}}(\boldsymbol{x})$ that unifies both objectives:

$$\min_{\theta} \mathcal{L}_{\boldsymbol{\theta}}(\boldsymbol{x}) = -L_{\boldsymbol{\theta}}(\boldsymbol{x} + \hat{\epsilon}(\boldsymbol{\theta}, \boldsymbol{x})) + \lambda L_{\boldsymbol{\theta}}(\boldsymbol{x}). \tag{6}$$

Thus, we only need to optimize $\mathcal{L}_{\boldsymbol{\theta}}(\boldsymbol{x})$ over natural image to achieve our objectives.

However, as we have not yet imposed any constraints on the values of the loss function, directly optimizing $\mathcal{L}_{\boldsymbol{\theta}}(\boldsymbol{x})$ results in non-convergence issue. To address this issue, we introduce an anchor model, leading to a dual-model framework. Namely, the anchor model is pre-trained to fit the natural image distribution, which is introduced to avoid the shift of the optimized overfitting model $\theta$.

$w(\cdot) : \mathbb{R}^d \to \mathbb{R}$ is an anchor model with fixed parameters, and $\theta(\cdot) : \mathbb{R}^d \to \mathbb{R}$ is an overfitting model with learnable parameters. The anchor model $w(\cdot)$ is a self-supervised model trained on real images, capable of encoding real images into consistent features. Then we train $\theta(\cdot)$ to achieve the overfitting objective. Specifically, we constrain the outputs of both $w(\cdot)$ and $\theta(\cdot)$ to be scalar values between 0 and 1, and define $L_{\boldsymbol{\theta}}(\boldsymbol{x}) = |w(\boldsymbol{x}) - \theta(\boldsymbol{x})|$. Naturally, this ensures that the minimum value of $L_{\boldsymbol{\theta}}(\boldsymbol{x})$ is 0, and the maximum value is 1. Substituting the result into Equation 6, we obtain our optimization objective:

$$\mathcal{L}_{\boldsymbol{\theta}}(\boldsymbol{x}) = -|w(\boldsymbol{x} + \hat{\epsilon}(\boldsymbol{\theta}, \boldsymbol{x})) - \theta(\boldsymbol{x} + \hat{\epsilon}(\boldsymbol{\theta}, \boldsymbol{x}))| + |w(\boldsymbol{x}) - \theta(\boldsymbol{x})|. \tag{7}$$

Considering Equation 4, $\hat{\epsilon}(\boldsymbol{\theta}, \boldsymbol{x})$ represents a vector in the neighborhood of $\mathbf{0}$ that points in the direction of $-\nabla_{\boldsymbol{x}} L_{\boldsymbol{\theta}}(\boldsymbol{x})$. Since $-\nabla_{\boldsymbol{x}} L_{\boldsymbol{\theta}}(\boldsymbol{x})$ is difficult to solve, but when the number of $\boldsymbol{x}$ is sufficiently large, $\hat{\epsilon}(\boldsymbol{\theta}, \boldsymbol{x})$ follows a Gaussian distribution, we sample it from a Gaussian distribution in our experiments. Under the above approximations, our loss function becomes:

$$\mathcal{L}_{\boldsymbol{\theta}}(\boldsymbol{x}) = -|w(\boldsymbol{x} + \boldsymbol{\epsilon}) - \theta(\boldsymbol{x} + \boldsymbol{\epsilon})| + |w(\boldsymbol{x}) - \theta(\boldsymbol{x})|. \tag{8}$$

After optimizing $L_{\boldsymbol{\theta}}(\boldsymbol{x})$ through training, the model produces a small value for $L(\boldsymbol{x})$ when a real image $\boldsymbol{x}$ is input, and a larger value for $L(\boldsymbol{x} + \boldsymbol{\epsilon})$. Since $\boldsymbol{x}$ represents a real image and $\boldsymbol{x} + \boldsymbol{\epsilon}$ represents a sample deviating from the real image (i.e., a generated image), we can utilize $L(\cdot)$ as a discriminator to determine whether an image is real.

### 3.4 DETAILS

Specifically, we extract features from the image $\boldsymbol{x}$ using the DINOv2 model (Oquab et al., 2024) to implement $w(\boldsymbol{x})$. For $\theta(\cdot)$, we first transform the image using two trainable convolutional layers into a vector of the same size as the original image. This transformed vector is then added to the original image, and the combined result is passed through DINOv2 to extract features, thereby implementing $\theta(\boldsymbol{x})$. This approach preserves the original feature extraction capability of the DINOv2 model, this will be discussed in detail in Section 4.3. Define the operations applied to the image prior to inputting it into the DINOv2 model as $g_{\boldsymbol{\theta}}(\cdot) : \mathbb{R}^d \to \mathbb{R}^d$, where the convolution operations are denoted as $c_{\boldsymbol{\theta}}(\cdot) : \mathbb{R}^d \to \mathbb{R}^d$, such that:

$$g(\boldsymbol{x}) = \boldsymbol{x} + \lambda_c c_{\boldsymbol{\theta}}(\boldsymbol{x}). \tag{9}$$

Denoting the DINOv2 model as $d(\cdot)$, we obtain:

$$w(\boldsymbol{x}) = d(\boldsymbol{x}), \theta(\boldsymbol{x}) = d(g_{\boldsymbol{\theta}}(\boldsymbol{x})). \tag{10}$$

Since the DINOv2 model outputs a feature vector, we extend the form of $L(\boldsymbol{x})$ to be the second-order norm, i.e., $L(\boldsymbol{x}) = ||w(\boldsymbol{x}) - \theta(\boldsymbol{x})||_2$. We visualize the overall process in Figure 1.

We employ validation-based early stopping by reserving 1,000 samples for validation purposes. Specifically, due to the relatively small number of trainable parameters, we evaluate the model's performance on the validation set after every 10 gradient descent iterations. The model checkpoint with the best validation performance is then selected for further evaluation on the test set.

## 4 EXPERIMENTS

In this section, we will first outline the experimental setups employed in our study, followed by a comprehensive presentation of the experimental results that substantiate the efficacy of our approach.

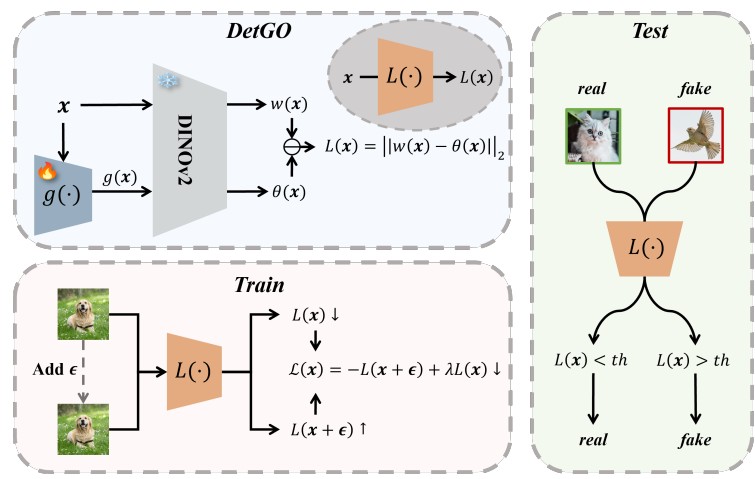

Figure 1: Framework of the proposed method during training and testing phases.

### 4.1 SETUP

**Training Datasets.** Unlike conventional detectors, DetGO is trained exclusively on real images, specifically utilizing the ImageNet dataset (Deng et al., 2009). We selected 100 images from each category, resulting in a total of 100,000 images.

**Testing Datasets.** To assess the generalization ability of our proposed method in practical contexts, we utilized a variety of real images, multiple GAN and diffusion models and several commercially available generative models following the work of (Stein et al., 2023a). For the real images, we utilized three datasets: ImageNet , LSUN-Bedroom (Yu et al., 2015), and LAION (Schuhmann et al., 2021). For the generated images, we selected outputs from a range of advanced generative models, including ADM (Dhariwal & Nichol, 2021), ADM-G, LDM (Rombach et al., 2022), DiT-XL2 (Peebles & Xie, 2023), BigGAN (Brock et al., 2019), GigaGAN (Kang et al., 2023), StyleGAN (Karras et al., 2019), RQ-Transformer (Lee et al., 2022), MaskGIT (Chang et al., 2022), DDPM (Ho et al., 2020), iDDPM (Nichol & Dhariwal, 2021), Diffusion Projected GAN (Wang et al., 2023b), Projected GAN and Unleasing Transformer (Bond-Taylor et al., 2022). Additionally, we conducted tests on GenImage (Zhu et al., 2023), a recently established benchmark for detecting AI-generated content. This benchmark includes a variety of models such as GLIDE (Nichol et al., 2022), VQDM (Gu et al., 2022), Stable Diffusion (Rombach et al., 2022), Wukong (Wukong, 2022), and Midjourney (Midjourney, 2022).

**Baselines.** We use both training methods and training-free methods as baselines. For training methods, we take CNNspot (Wang et al., 2020), Ojha (Ojha et al., 2023), DIRE (Wang et al., 2023a), and NPR (Tan et al., 2023a) as baselines. For training-free methods, we take AEROBLADE (Ricker et al., 2024) as baselines. On GenImage, we also report the result of F3Net (Qian et al., 2020), GANDetection (Mandelli et al., 2022), LGrad (Tan et al., 2023b), ResNet-50 (He et al., 2016), DeiT-S (Touvron et al., 2021), Swin-T (Liu et al., 2021), Spec (Zhang et al., 2019) and GramNet (Liu et al., 2020).

**Experiment details.** Specifically, to balance detection performance and efficiency, we use DINOv2-ViT-L/14, which will be discussed in 4.3. During the training and testing phases, the images fed into the network undergo random cropping to a size of $224 \times 224$ pixels, and all images are in PNG format. We utilize the Stochastic Gradient Descent (SGD) optimizer with a batch size of 32 and a learning rate of 0.01. Additionally, we implement early stopping to ensure optimal performance during training. To evaluate the performance of the proposed method, we adopt the metrics used in the baseline studies, which include the Area Under the Receiver Operating Characteristic curve

Table 1: Fake image detection performance on ImageNet. Values are percentages. **Bold** numbers are superior results. A higher value indicates better performance.

| Methods | ADM | | ADMG | | LDM | | DiT | | Models BigGAN | | GigaGAN | | StyleGAN XL | | RQ-Transformer | | Mask GIT | | Average | |
|---|---|---|---|---|---|---|---|---|---|---|---|---|---|---|---|---|---|---|---|---|
| | AUROC | AP | AUROC | AP | AUROC | AP | AUROC | AP | AUROC | AP | AUROC | AP | AUROC | AP | AUROC | AP | AUROC | AP | AUROC | AP |
| AEROBLADE | 50.49 | 50.24 | 57.27 | 56.57 | 61.02 | 57.50 | 71.54 | 71.40 | 50.14 | 51.74 | 55.50 | 53.90 | 50.56 | 52.42 | 69.33 | 68.48 | 58.08 | 57.28 | 58.21 | 57.73 |
| CNNspot | 71.25 | 68.21 | 70.27 | 66.23 | 70.34 | 64.41 | 53.02 | 48.86 | 86.11 | 81.98 | 66.96 | 63.61 | 68.28 | 64.37 | 60.12 | 57.21 | 73.86 | 68.22 | 68.91 | 64.79 |
| Ojha | 83.24 | 83.28 | 77.48 | 76.32 | 83.23 | 82.66 | **80.00** | 78.10 | 90.70 | 89.46 | 81.33 | 79.23 | 81.97 | 79.28 | 82.61 | 80.71 | 84.63 | 86.07 | 82.80 | 81.68 |
| DIRE | 57.82 | 58.57 | 55.95 | 53.84 | 57.59 | 58.62 | 50.38 | 51.99 | 50.46 | 50.10 | 49.16 | 52.42 | 51.99 | 53.36 | 51.80 | 50.45 | 49.74 | 50.01 | 52.76 | 53.26 |
| NPR | 76.68 | 74.10 | 77.26 | 74.49 | **92.73** | **88.74** | 79.44 | 73.18 | 81.48 | 78.55 | 80.22 | 77.07 | 80.91 | 77.52 | 86.49 | 83.55 | **89.75** | 86.32 | 82.77 | 79.27 |
| DetGO | **86.09** | **85.74** | **79.30** | **78.73** | 73.41 | 84.09 | 70.79 | **82.72** | **91.03** | **90.50** | **87.26** | **92.53** | **88.49** | **93.10** | **88.23** | **93.17** | 82.90 | **89.87** | **83.06** | **87.82** |

Table 2: Fake image detection performance on GenImage. Except for DetGO, all methods require training on Stable Diffusion V1.4.

| Methods | Models | | | | | | Average |
|---|---|---|---|---|---|---|---|
| | Midjourney | ADM | GLIDE | Wukong | VQDM | BigGAN | |
| ResNet-50 | 54.90 | 53.50 | 61.90 | 98.20 | 56.60 | 52.00 | 62.85 |
| DeiT-S | 55.60 | 49.80 | 58.10 | 98.90 | 56.90 | 53.50 | 62.13 |
| Swin-T | 62.10 | 49.80 | 67.60 | 99.10 | 62.30 | 57.60 | 66.42 |
| CNNDet | 52.80 | 50.10 | 39.80 | 78.60 | 53.40 | 46.80 | 53.58 |
| Spec | 52.00 | 49.70 | 49.80 | 94.80 | 55.60 | 49.80 | 58.62 |
| F3Net | 50.10 | 49.90 | 50.00 | **99.90** | 49.90 | 49.90 | 58.28 |
| GramNet | 54.20 | 50.30 | 54.60 | 98.90 | 50.80 | 51.70 | 60.08 |
| DIRE | 60.20 | 50.90 | 55.00 | 99.20 | 50.10 | 50.20 | 60.93 |
| Ojha | **73.20** | 55.20 | **76.90** | 75.60 | 56.90 | 80.30 | 69.68 |
| DetGO | 70.66 | **71.99** | 70.96 | 69.10 | **82.93** | **88.06** | **75.61** |

(AUROC, AUC), average precision score (AP), and accuracy (ACC). Due to the extensive size of GenImage and the time-consuming nature of certain detection methods, we opted to directly utilize the scores reported by certain baselines as presented in the corresponding articles.

## 4.2 RESULTS

**Comparison to Existing Detectors.** Given that DetGO was trained on the ImageNet dataset, we initially utilized ImageNet as the real-image dataset to compare the performance of our approach with various baselines. Following the methodology outlined in (Stein et al., 2023b), our experimental results on the ImageNet dataset are presented in Table 1. The generative models presented are all trained on the ImageNet dataset. It is evident that DetGO effectively distinguishes between real and generated images, and demonstrates consistent performance across various generative models, and outperforms all compared methods. Furthermore, we tested the performance of DetGO on the GenImage dataset. In these tests, real image dataset is still ImageNet and the results of the compared baselines are sourced from the GenImage paper and were obtained using models trained with Stable Diffusion V1.4. Since Stable Diffusion V1.4 and v1.5 are too similar and all baselines achieve an AUROC of 99, we exclude this set of data from our results. Aside from Stable Diffusion V1.5, these baselines exhibit a significant decline in performance on datasets that were not encountered during training. In contrast, DetGO demonstrated consistent performance with the highest average accuracy.

**Generalization Capability Evaluation.** Unlike the previous context, when assessing the generalization capability of DetGO, the real and generated images used were unseen by the detector during training, while no such restrictions were imposed on the baselines. Table 3 displays the detection performance on real images from the LSUB-bedroom and generated images produced by models trained on the LSUB-bedroom dataset. DetGO demonstrated superior performance compared other models. To address the challenges posed by the rising prevalence of video generation models to digital security, we also tested our model's performance on generated video frames. Our experimental setup was based on the recently prominent Sora model (OpenAI, 2024). Since Sora is not publicly available, we utilized several demonstration videos from the official Sora website. Specifically, we selected 50 publicly available videos from Sora and extracted $5,000$ frames each to compile our dataset. For real images, we chose $5,000$ pictures from the LAION-400m dataset. Table 4 presents

Table 3: Fake image detection performance on LSUN-BEDROOM.

| Methods | Models | | | | | | | | | | | | | | |
| | ADM | | DDPM | | iDDPM | | Diffusion GAN | | Projected GAN | | StyleGAN | | Unleashing Transformer | | Average | |
| | AUROC | AP | AUROC | AP | AUROC | AP | AUROC | AP | AUROC | AP | AUROC | AP | AUROC | AP | AUROC | AP |
|---|---|---|---|---|---|---|---|---|---|---|---|---|---|---|---|---|
| AEROBLADE | 55.96 | 58.62 | 70.67 | 71.71 | 69.64 | 67.69 | 49.05 | 50.53 | 52.47 | 49.79 | 49.68 | 51.41 | 56.43 | 57.00 | 57.70 | 58.11 |
| CNNspot | 65.97 | 63.55 | 75.53 | 72.91 | 76.37 | 73.89 | 82.80 | 83.16 | 85.42 | 85.47 | **98.36** | **98.42** | 91.58 | 91.43 | 82.29 | 81.26 |
| Ojha | **71.52** | 70.72 | 80.52 | 79.89 | 79.88 | 79.36 | 86.08 | 84.22 | 86.91 | 85.51 | 83.75 | 82.86 | 85.86 | 84.97 | 82.08 | 81.01 |
| DIRE | 54.47 | 56.39 | 57.30 | 62.34 | 59.08 | 61.47 | 54.16 | 56.79 | 55.18 | 55.11 | 57.90 | 56.98 | 61.69 | 64.77 | 57.11 | 59.12 |
| NPR | 68.70 | 63.81 | 82.97 | 75.63 | 71.72 | 66.62 | 81.77 | 73.94 | 83.56 | 75.82 | 65.33 | 58.78 | 80.14 | 72.35 | 76.31 | 69.56 |
| DetGO | 71.23 | **71.43** | **85.77** | **86.31** | **83.06** | **83.40** | **91.21** | **90.93** | **91.84** | **91.61** | 80.14 | 81.51 | **92.22** | **92.17** | **85.07** | **85.33** |

Table 4: Fake image detection performance on Sora.

| Model | Methods | | | | | | | | | | | |
| | CNNspot | | Ojha | | NPR | | DIRE | | AEROBLADE | | DetGO | |
| | AUROC | AP | AUROC | AP | AUROC | AP | AUROC | AP | AUROC | AP | AUROC | AP |
|---|---|---|---|---|---|---|---|---|---|---|---|---|
| Sora | 59.58 | 56.36 | 73.64 | 74.17 | 78.07 | 61.43 | 60.20 | 56.05 | 62.37 | 63.48 | **87.64** | **88.07** |

our results, demonstrating that DetGO also achieved the best performance on novel datasets. The experiments highlighted above demonstrate that our model exhibits strong generalization capabilities across various generative models and datasets.

**Robustness to post-processing operations.** In real-world scenarios, images are seldom pristine; they undergo continuous compression and interference during dissemination on social media. Detection models that perform well on clean images may experience diminished performance on distorted ones. In this section, we evaluate the robustness of DetGO against interference. We introduce disturbances at five levels of Gaussian blur ($\sigma = 1, 2, 3, 4, 5$), Gaussian noise ($\lambda = 0.05, 0.1, 0.15, 0.2, 0.25$), and JPEG compression (quality: q = 90, 80, 70, 60, 50). We explore the robustness of the previously well-performing baseline: CNNSpot, Ojha, NPR and DetGO. The results are presented in Figure 2. As shown in the results, DetGO exhibits the best robustness when faced with degraded images. The feature extraction method based on pixel relationships, NRP, experienced significant degradation. In contrast, our approach leverages the strong generalization capability of DINOv2, achieving superior results across various interference tests, thereby demonstrating its effectiveness in real-world applications.

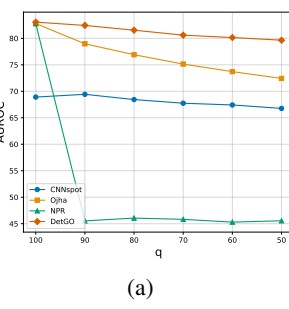 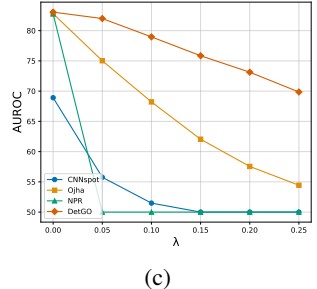

(a)  (b)  (c)

Figure 2: Robustness to post-processing operations. (a) shows the robustness to JPEG compression, (b) shows the robustness to Gaussian blur, and (c) shows the robustness to Gaussian noise.

**Robustness to data transformations.** Robustness to data transformations is an essential property for models to maintain consistent performance under a range of perturbations that may occur in real-world scenarios. When applied to the ImageNet dataset, for instance, real images that undergo common transformations—such as random cropping, resizing, rotation, or color jittering—may be misclassified by models as fake or synthetic. This phenomenon arises because these transformations, which are typically employed during model training for data augmentation, introduce subtle perturbations that can alter the distribution of pixel values and higher-level features. Consequently, a model trained to distinguish between real and fake images may misinterpret these legitimate variations as artifacts indicative of synthetic generation. This sensitivity suggests that the decision boundaries

Table 5: Fake image detection performance on ImageNet with data transformations.

| Transformation | Intensity | ADM | ADMG | LDM | DiT | BigGAN | GigaGAN | StyleGAN XL | RQ-Transformer | Mask GIT | Average |
|---|---|---|---|---|---|---|---|---|---|---|---|
| None | | 86.09 | 79.30 | 73.41 | 70.79 | 91.03 | 87.26 | 88.49 | 88.23 | 82.90 | 83.06 |
| Rotation | $-45° \sim 45°$ | 83.05 | 76.94 | 65.32 | 67.95 | 85.88 | 83.01 | 84.12 | 83.24 | 75.26 | 78.31 |
| | $-90° \sim 90°$ | 80.57 | 74.98 | 65.50 | 66.64 | 83.88 | 79.93 | 81.11 | 81.18 | 72.81 | 76.29 |
| Flip | horizontal $p = 0.5$ | 86.15 | 79.39 | 70.29 | 71.89 | 90.99 | 87.93 | 88.68 | 87.84 | 81.16 | 82.70 |
| | vertical $p = 0.5$ | 82.31 | 76.51 | 68.94 | 69.29 | 87.74 | 82.78 | 83.63 | 83.10 | 77.79 | 79.12 |
| Brightness jitter | $-0.25 \sim 0.25$ | 85.92 | 79.29 | 70.05 | 71.12 | 90.35 | 87.67 | 88.45 | 87.83 | 81.11 | 82.42 |
| | $-0.5 \sim 0.5$ | 85.21 | 79.28 | 69.92 | 71.13 | 90.16 | 86.67 | 88.10 | 87.69 | 80.64 | 82.09 |
| Contrast jitter | $-0.25 \sim 0.25$ | 86.04 | 79.15 | 70.01 | 71.45 | 90.68 | 87.79 | 88.38 | 88.03 | 80.80 | 82.48 |
| | $-0.5 \sim 0.5$ | 85.54 | 79.24 | 69.57 | 71.18 | 90.17 | 86.61 | 87.83 | 87.37 | 80.77 | 82.03 |
| Saturation jitter | $-0.25 \sim 0.25$ | 86.19 | 79.24 | 70.15 | 71.80 | 90.76 | 88.01 | 88.77 | 88.02 | 81.19 | 82.68 |
| | $-0.5 \sim 0.5$ | 85.86 | 79.22 | 70.45 | 71.23 | 90.53 | 87.56 | 88.81 | 87.82 | 80.92 | 82.49 |
| Hue jitter | $-0.25 \sim 0.25$ | 85.41 | 78.34 | 69.95 | 71.89 | 88.04 | 85.50 | 87.91 | 86.71 | 81.16 | 81.66 |
| | $-0.5 \sim 0.5$ | 83.80 | 78.28 | 69.04 | 70.29 | 87.45 | 85.15 | 87.55 | 85.52 | 80.32 | 80.82 |

learned by the model might be overly reliant on superficial characteristics, rather than capturing the fundamental semantic content of the images. Our results, as shown in Table 5, demonstrate that the proposed method exhibits relative robustness to various data augmentations. This robustness can be attributed to the DINOv2 model's extensive pretraining on large-scale real-world image datasets, which equips it with a strong capability to capture invariant features under different transformations. Consequently, the model can maintain stable performance when subjected to natural variations in real images. We only observed a relatively significant performance drop under random rotation transformations. This decline can likely be attributed to the pixel interpolation process introduced during rotation, which may cause a loss of fine-grained details in the image.

## 4.3 ABLATION STUDY

This section examines the effects of models, convolutional layers, training perturbations, and early stopping on detection performance. We found that smaller models like DINOv2-S/14 significantly underperformed. We set the dimensionality of convolutional layers to 1 for efficiency, as it had minimal impact on results. Adding Gaussian noise $\epsilon$ showed that both low and high perturbation levels hindered generalization. Our early stopping strategy also revealed that optimal test performance did not occur at minimum loss, highlighting the importance of validation. These insights underscore key factors influencing detection effectiveness. Additionally, we investigated the impact of placing trainable layers either before the input or after the output of the DINOv2 model. Our results show that the latter configuration tends to degrade DINOv2's feature extraction capabilities.

**The effect of models.** In our experiments, we primarily utilized the DINOv2-ViT-L/14 model. This section explores the impact of different DINOv2 model sizes on performance. The results, as shown in Table 6, indicate that the performance of ViT-L/14 and ViT-g/14 is similar, while a noticeable performance drop occurs with the smaller ViT-S/14 and ViT-B/14 models. This decline may be attributed to the smaller models' inability to effectively capture the differences between real and generated images. Considering detection efficiency, we opted for the more balanced ViT-L/14 model in our experiments.

**The effect of convolutional layer.** In "Details" 3.4 of the Method section, we provide a comprehensive elaboration on the structural composition of the function $\theta(\cdot)$. This discussion further explores the implications of the dimensionality of the intermediate layer within the convolutional network framework and the convolutional coefficients $\lambda_c$ in the $g(\cdot)$ function. Table 7 and 8 illustrates the impact of these factors on detection performance. It can be observed that the dimensionality of the intermediate layers in the convolutional network has a negligible impact on the final detection performance. Therefore, to expedite training, we set the dimensionality of the intermediate layers to 1. We next focus on the effect of the convolutional coefficient. When the coefficient is set to a very low value, the function $g$ fails to introduce meaningful changes to the image $x$, rendering the subsequent feature extraction model ineffective at distinguishing $x$ from $g(x)$. Conversely, when the coefficient is too large, it overly distorts $x$, thereby obscuring the differences in distinguishing characteristics between the real and generated images, i.e., the differences between $x$ and $g(x)$.

**The effect of training perturbations.** In the training process, we introduce Gaussian noise $\epsilon$ to the original image $x$. Our experimental results demonstrate that variations in the intensity of this noise significantly influence the detection performance of the model, as shown in Table 9. When the perturbation is minimal, the model tends to overfit to the training set rather than learning meaningful representations of the real images. Conversely, when the perturbation is excessively large, the model only learns to distinguish between real images and pure noise, which also leads to a deterioration in detection performance. This trade-off indicates that an optimal level of perturbation is crucial for effective model training, as it ensures that the model captures the inherent characteristics of the data.

Table 6: The effect of the size of DINOv2.

| Size | ViT-S/14 | ViT-B/14 | ViT-L/14 | ViT-g/14 |
|---|---|---|---|---|
| AUC | 63.01 | 73.43 | 83.06 | 80.56 |

Table 7: The effect of the intermediate layer dimension.

| Dim | 1 | 2 | 3 | 4 | 5 |
|---|---|---|---|---|---|
| AUC | 83.06 | 82.01 | 82.23 | 81.22 | 80.83 |

Table 8: The effect of convolutional coefficients.

| $\lambda_c$ | 0.05 | 0.1 | 0.15 | 0.2 | 0.25 | 0.3 | 0.4 | 0.5 |
|---|---|---|---|---|---|---|---|---|
| AUC | 52.03 | 74.69 | 83.06 | 82.88 | 82.61 | 78.79 | 73.27 | 66.86 |

Table 9: The effect of noise intensity during training.

| $\lambda_n$ | 0.1 | 0.2 | 0.3 | 0.4 | 0.5 | 0.6 | 0.7 |
|---|---|---|---|---|---|---|---|
| AUC | 73.97 | 78.69 | 83.06 | 82.44 | 80.12 | 74.15 | 68.18 |

Table 10: The effect of loss weight.

| $\lambda$ | 0.02 | 0.05 | 0.1 | 0.2 | 0.3 | 0.5 |
|---|---|---|---|---|---|---|
| AUC | 71.35 | 80.94 | 83.06 | 82.17 | 80.23 | 76.51 |

**The effect of loss weight.** In the Equation 8, we introduced our proposed loss function, which is formulated as a weighted $\ell_2$-norm of the difference between two feature vectors. This formulation includes a hyperparameter, $\lambda$, that controls the relative importance of the feature difference term in the overall loss. In this section, we investigate the effect of varying the $\lambda$ value on the model's performance. The choice of $\lambda$ significantly influences the learning dynamics, as it governs the sensitivity of the model to discrepancies in the feature representations. When $\lambda$ is too small, the loss function may not adequately penalize deviations between features, potentially leading to a model that underfits and fails to capture subtle distinctions between real and synthetic images. Conversely, an excessively large $\lambda$ value could dominate the learning process, causing the model to prioritize minimizing feature differences at the expense of other critical loss components, thereby hindering its ability to generalize. Our results, as depicted in Table 10, show that there is an optimal range for $\lambda$, where the model achieves a balance between feature alignment and overall classification performance.

**Validation-based early stopping.** As mentioned previously in 3.4, we evaluate the model's performance on the validation set after every 10 gradient descent iterations to select the optimal checkpoint. In Figure 4, we present how the performance on both the validation and test sets evolves throughout the training process. During the training process, the loss continues to decrease; however, it is clear that the model does not achieve higher performance at lower loss values. Our early stopping strategy effectively ensures optimal performance on the test set.

**Trainable layer placement.** In our experiments, we initially positioned the trainable convolutional layers between the input images and the DINOv2 model. In this section, we investigate the effects of relocating the trainable layers to follow the DINOv2 model, utilizing linear layers as the trainable components. We refer to

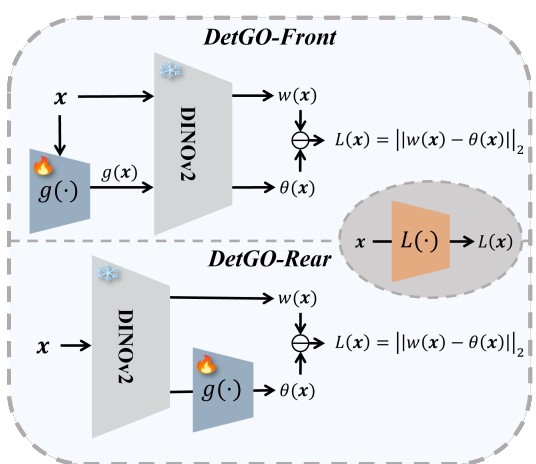

Figure 3: Pipeline of DetGO-Rear.

Table 11: Comparison of detection performance between DetGO and its variants on ImageNet.

| Methods | Models | | | | | | | | | Average |
| --- | ADM | ADMG | LDM | DiT | BigGAN | GigaGAN | StyleGAN XL | RQ-Transformer | Mask GIT | |
| DetGO-Front | 86.09 | 79.30 | 73.41 | 70.79 | 91.03 | 87.26 | 88.49 | 88.23 | 82.90 | 83.06 |
| DetGO-Rear | 79.31 | 71.93 | 67.11 | 69.15 | 89.31 | 82.15 | 87.97 | 84.65 | 82.73 | 79.37 |

this variant as DetGO-Rear, while the original method is designated as DetGO-Front for comparison. The detection architecture is illustrated in Figure 3, while the performance results on the ImageNet dataset are summarized in Table 11. Our findings reveal that when the trainable layers are placed after the DINOv2 model, the overall performance is inferior compared to the configuration where these layers precede the DINOv2. This decline in performance suggests that placing trainable layers after DINOv2 may compromise the model's inherent feature extraction capabilities. DINOv2 is designed to capture rich, high-level representations from the input data, and the introduction of trainable layers after it's output layer can interfere with the effective utilization of these features, thereby diminishing the model's ability to leverage the discriminative features that are critical for accurate classification. This ablation study underscores the importance of strategically positioning trainable components within the network architecture to preserve the integrity of the feature extraction process and enhance overall model performance.

## 5 LIMITATION

The success of DetGO hinges on the availability of high-quality real image datasets for training. While our approach does not require synthetic images, it necessitates extensive collections of real-world images that are both diverse and representative of the natural image distribution. This dependency could pose a challenge in domains where access to high-quality, unbiased real images is limited or constrained by privacy concerns.

The introduction of two separate models—the anchor model and the overfitting model—significantly increases the computational overhead, particularly during the training phase. This dual-model structure requires additional memory and training time compared to traditional single-model approaches. Although it can be mitigated by employing smaller versions of DINOv2, optimizing computational efficiency remains an open challenge for large-scale deployments or real-time applications.

## 6 CONCLUSION

In this paper, we introduced DetGO, a novel approach for detecting AI-generated images by leveraging over-fitting to the distribution of real images. Unlike conventional detection methods that rely on the existence of generation-specific artifacts or require access to synthetic examples during training, DetGO capitalizes on the inherent distributional differences between real and generated images. Through a dual-model framework comprising an anchor model and an overfitting model, DetGO ef-

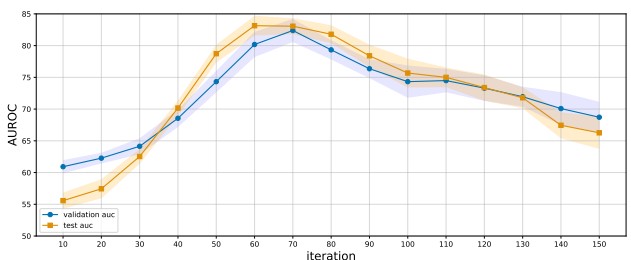

Figure 4: Performance throughout the training process.

fectively highlights the mismatch in loss landscapes, achieving state-of-the-art detection performance across multiple benchmarks, including various GANs, diffusion models, and commercially available generative models. Our extensive experimental evaluation demonstrates that DetGO not only outperforms existing detectors on standard datasets but also maintains robust generalization capability in the face of increasingly sophisticated generative models. Additionally, DetGO exhibits resilience against typical post-processing operations, making it a promising candidate for real-world deployment scenarios.

## ETHICS STATEMENT

We affirm that this research adheres to the ICLR Code of Ethics. This work does not involve the use of human subjects, private data, or datasets with sensitive or restricted content. All experiments were conducted using publicly available image datasets.

The proposed method, DetGO, is intended solely for detecting AI-generated images and does not contribute to the generation or dissemination of misleading or harmful content. The study is designed to address potential misuse of AI technology by enhancing detection capabilities and does not introduce risks or adverse consequences that could result from its implementation.

The methodology and results are reported with full transparency to ensure reproducibility and ethical standards in research. The work complies with all applicable legal and ethical guidelines, and the authors have no conflicts of interest or sponsorships that could influence the research outcomes. Additionally, all references and related works have been appropriately cited, and the research upholds the principles of academic integrity and responsible conduct of research.

## REPRODUCIBILITY STATEMENT

To ensure the reproducibility of our results, we have provided detailed descriptions of the experimental setup, model architecture, and training procedures in Section 4. Specifically, the training datasets, hyperparameters, and evaluation metrics are clearly outlined to facilitate replication. All models were trained using publicly available datasets and we specify the data preprocessing steps employed in Section 4.1. All datasets required for our experiments can be directly accessed through the links or official websites specified in the references (Stein et al., 2023a; Zhu et al., 2023; Schuhmann et al., 2021; OpenAI, 2024).

The proposed method, DetGO, is implemented using PyTorch, and the architecture details, including the dual-model framework and loss functions, are provided in Sections 3.2 and 3.3. Code will be available once the paper is accepted.

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
