# OpenReview forum: "Overfitting: An Unexpected Asset in AI‐Generated Image Detection"
_ICLR.cc/2025/Conference — Submitted to ICLR 2025_

### Official Review · Reviewer_dZhV · 2024-10-31

**Soundness:** 2
**Presentation:** 3
**Contribution:** 2
**Rating:** 6
**Confidence:** 3

**Summary:**

This paper propose a SAM-based approach to detect generated images by overfitting the distribution of natural images. Additionally, an anchor model are utilized to solve the divergence issue. The authors conduct extensive experiments across multiple AI-image benchmarks and the results demonstrate the effectiveness of the approach.

**Strengths:**

1. This paper is well-written and easy to follow. The structure is clear.
2. The motivation of this paper is similar to Out Of Distribute (OOD) detection, where the model is only trained and fitted on real images. However, detecting generated images by overfitting is enlightening.
3. The methodology is logical, and the motivation is intuitive and profound. The design of sharpening the first derivative of the loss is intriguing.
4. Extensive and effective experiments proving the effectiveness of the proposed method. Ablation experiments validate the hypotheses.

**Weaknesses:**

1. Some technical or motivational clarification is needed.
2. Some ablation studies are recommended.
Please refer to Questions.

**Questions:**

1.  In line 196, the authors believe that ε follows a Gaussian distribution, I wonder whether there is a reasonable explanation.
2. Meanwhile, I believe that when ε is designed to be sampled from a Gaussian distribution, the proposed approach will suffer from performance degradation when Gaussian blur is applied to images, as the added noise ε’ and the designed ε belong to the same distribution, resulting in an increase in L(x+ε’). I hope the authors can clarify some of the points.
3. The AUROC and AP metrics are insensitive to classification thresholds. Based on this, they may not fully reflect the generalization ability of the model. I hope the authors can provide more experimental results, such as the AUROC and AP on real samples of LSUN-Bedroom (Table 3) and generated samples from ImageNet settings (Table 1), or the Accuracy with fixed thresholds on different datasets.
4. Why is g(x) designed as multiple trainable convolutional layers? Would other architectures, such as U-Net, yield better or worse results?
5. Will the selection of the training dataset affect the detection performance?
6. Can you evaluate the complexity of the proposed DetGO?

Based on the above Strengths and Weaknesses, I will give a Borderline Score. I will revise my score if the author's rebuttal can provide reasonable explanations.

---

> ### Author Response · Authors · 2024-11-22
> **PART 1**
>
> We would like to sincerely thank the reviewer for taking the time to review our manuscript! In response to the 6 questions raised by the reviewer, we provide the following explanations in turn:
>
> >Q.1: In line 196, the authors believe that ε follows a Gaussian distribution, I wonder whether there is a reasonable explanation.
>
> Thanks for pointing out this potentially confusing explanation. $\hat{\epsilon}$ represents the vector that minimizes the inner product with the gradient of $L$ in image space under the $\ell_2$ norm constraint (as shown in Equation 4). The choice of $\hat{\epsilon}$ is specific to $L$, which is related to the design of our network, specifically to the implementation of the functions $w$ and $\theta$. Conceptually, $\epsilon$ points from the sample $x$ towards a nearby minimum of $L$. Finding this value is challenging, so we resort to random sampling near $x$ in an attempt to cover this minimum. A practical approach to this is random sampling from a normal distribution, which is why we model $\epsilon$ as Gaussian noise.
>
> When using noise sampled from a uniform or laplace distribution, comparable results are obtained. Under the experimental setup in Table 1, the average AUROC/AP performance is as follows:
>
> | MODELS | Gaussian noise | Uniform noise | Laplace noise |
> | - | - | - | - |
> | | AUROC/AP | AUROC/AP | AUROC/AP |
> | ADM | 86.09/85.74 | 86.14/87.33 | 84.57/83.94 |
> | ADMG | 79.30/78.73 | 78.90/79.72 | 78.85/78.72 |
> | LDM | 73.41/84.09 | 68.96/70.07 | 67.47/77.12 |
> | DiT | 70.79/82.72 | 68.92/71.12 | 69.23/80.34 |
> | BigGAN | 91.03/90.50 | 91.26/91.73 | 90.17/88.85 |
> | GigaGAN | 87.26/92.53 | 86.88/86.86 | 85.36/83.70 |
> | StyleGAN XL | 88.49/93.10 | 88.73/88.64 | 87.31/84.91 |
> | RQ-Transformer | 88.23/93.17 | 87.88/87.92 | 87.47/87.34 |
> | Mask GIT | 82.90/89.87 | 81.56/81.74 | 81.63/85.89 |
> | Average | 83.06/87.82 | 82.13/82.79 | 81.34/83.42 |
>
> In response to your valuable question, we will add the above explanations, results, and discussions to the revision.
>
> >Q.2: Meanwhile, I believe that when ε is designed to be sampled from a Gaussian distribution, the proposed approach will suffer from performance degradation when Gaussian blur is applied to images, as the added noise ε’ and the designed ε belong to the same distribution, resulting in an increase in L(x+ε’). I hope the authors can clarify some of the points.
>
> A.2: We agree with your point. Moreover, our experimental results shown in panels (b) and (c) of Figure 2 provides similar conclusions, where Gaussian blur or Gaussian noise is applied to the images during the testing phase. Namely, for all tested detection methods, performance degrades in the presence of such adversarial perturbations. We would like to highlight that our method demonstrates more robust detection performance. We will highlight the results and conclusions in our revision.
>
> >Q.3: The AUROC and AP metrics are insensitive to classification thresholds. Based on this, they may not fully reflect the generalization ability of the model. I hope the authors can provide more experimental results, such as the AUROC and AP on real samples of LSUN-Bedroom (Table 3) and generated samples from ImageNet settings (Table 1), or the Accuracy with fixed thresholds on different datasets.
>
> Thanks for your constructive suggestion. Below, we present the AUC/AP results for the real-swapped datasets from Tables 1 and 3, r for real and f for fake. The results obtained are comparable with the original setup, demonstrating the robustness of our method. In comparison, the experimental group that includes ImageNet (real) exhibits a slightly higher proportion of better performance, which may be attributed to the fact that our convolutional layers were trained on ImageNet.
>
> | MODELS | ImageNet(r) ImageNet(f) | LSUN(r) ImageNet(f) |
> | - | - | - |
> | ADM | 86.09/85.74 | 83.86/86.86 |
> | ADMG | 79.30/78.73 | 76.30/81.25 |
> | LDM | 73.41/84.09 | 65.24/72.69 |
> | DiT | 70.79/82.72 | 62.90/70.72 |
> | BigGAN | 91.03/90.50 | 90.50/92.58 |
> | GigaGAN | 87.26/92.53 | 85.33/87.59 |
> | StyleGAN XL | 88.49/93.10 | 87.02/88.95 |
> | RQ-Transformer | 88.23/93.17 | 88.66/90.62 |
> | Mask GIT | 82.90/89.87 | 79.18/83.14 |
> | Average | 83.06/87.82 | 79.89/83.82 |
>
> | MODELS | LSUN(r)-LSUN(f) | ImageNet(r) LSUN(f) |
> | - | - | - |
> | ADM | 71.23/71.43 | 74.73/70.38 |
> | DDPM | 85.77/86.31 | 88.09/85.96 |
> | iDDPM | 83.06/83.40 | 85.20/82.39 |
> | Diffusion GAN | 91.21/90.93 | 91.70/89.97 |
> | Projected GAN | 91.84/91.61 | 91.91/89.59 |
> | StyleGAN | 80.14/81.51 | 83.24/81.59 |
> | Unleashing Transformer | 92.22/92.17 | 91.77/89.56 |
> | Average | 85.07/85.33 | 86.66/84.21 |
>
> The following presents the test set accuracy performance using the optimal average accuracy threshold obtained from the validation set.
>
> | MODELS | ACC |
> | - | - |
> | ADM | 80.97 |
> | ADMG | 74.33 |
> | LDM | 67.35 |
> | DiT | 68.67 |
> | BigGAN | 84.89 |
> | GigaGAN | 81.92 |
> | StyleGAN XL | 83.85 |
> | RQ-Transformer | 83.75 |
> | Mask GIT | 77.80 |
> | Average | 78.17 |

---

> ### Author Response · Authors · 2024-11-22
> **PART 2**
>
> >Q.4: Why is g(x) designed as multiple trainable convolutional layers? Would other architectures, such as U-Net, yield better or worse results?
>
> A.4: The use of two convolutional layers in $ g_\theta(x) $ is intended to provide a lightweight transformation. The first layer captures basic patterns and edges, while the second layer refines these features. This refined output is then added to the original image with a small coefficient, allowing for subtle adjustments to the input image without significantly altering its structure. This design ensures that the transformed output remains close to the original image, which is essential for preserving the integrity of DINOv2's feature extraction while allowing for minor perturbations.
>
> Furthermore, due to the small coefficient of the transformation, the desired effect is achieved with this lightweight structure alone. In the experiment, using wider networks with the same architecture does not lead to performance improvement (Table 7); instead, it increases the training cost. When more advanced architectures, such as U-Net, are employed, a similar performance is also observed:
>
> | MODELS | DetGO | DetGO-unet |
> | - | - | - |
> | | AUROC/AP | AUROC/AP |
> | ADM | 86.09/85.74 | 87.98/87.98 |
> | ADMG | 79.30/78.73 | 81.69/81.64 |
> | LDM | 73.41/84.09 | 72.95/71.66 |
> | DiT | 70.79/82.72 | 72.91/72.47 |
> | BigGAN | 91.03/90.50 | 91.77/90.71 |
> | GigaGAN | 87.26/92.53 | 88.59/87.44 |
> | StyleGAN XL | 88.49/93.10 | 90.23/88.33 |
> | RQ-Transformer | 88.23/93.17 | 90.05/88.33 |
> | Mask GIT | 82.90/89.87 | 84.78/83.57 |
> | Average | 83.06/87.82 | 84.55/83.57 |
>
> Thanks again for your insightful question. We will add the results and discussions to the revision.
>
> >Q.5: Will the selection of the training dataset affect the detection performance?
>
> A.5: Using real-world image datasets with richer scenes and content tends to yield better results. The results presented in the paper reflect the performance of the network trained on ImageNet. When we used the LSUN-bedroom dataset, we observed relatively worse performance. To ensure fairness, we present the experimental results for both settings on the LAION-Sora detection task.
>
> | imagenet | lsun |
> | - | - |
> | AUROC/AP | AUROC/AP |
> | 87.64/88.07 | 83.85/86.96 |
>
> When we use datasets with a limited range of scenes, the convolutional layers in our trained network may capture more specialized scene information, rather than generalizable features from real-world images.
>
> >Q.6: Can you evaluate the complexity of the proposed DetGO?
>
> A.6: Compared to the Transformer backbone used in DINOv2, the complexity of our lightweight, trainable convolutional layers is minimal. Additionally, we use a dual-model structure when evaluating a single image, so our method requires two forward passes for detection. The complexity of a single model is evaluated as follows: 77.82G FLOPs, 302.91M parameters.

---

> ### Comment · Reviewer_dZhV · 2024-11-24
>
> Thank you for your rebuttal, which addresses most of my concerns. However, I still have some questions.
> The author does not seem to have discussed that the proposed method has poor robustness to Gaussian noise. Meanwhile, I wonder if the classification of the method depends on the level of Gaussian noise, for example, when real images are added with stronger noise while fake images are weaker, will the method make a misjudgment?

---

> ### Author Response · Authors · 2024-11-24
>
> > The author does not seem to have discussed that the proposed method has poor robustness to Gaussian noise.
>
> Thank you for raising this point. We acknowledge that our method is not robust to Gaussian noise, as it leads to performance degradation, likely because the added noise $\epsilon$ and the designed noise belong to the same distribution, resulting in an increase in $L(x+\epsilon)$. We will further highlight this observation in the revised manuscript. However, as shown in Figure 2(c), we would like to emphasize that our method demonstrates relatively superior performance under such perturbations.
>
> > I wonder if the classification of the method depends on the level of  Gaussian noise, for example, when real images are added with stronger  noise while fake images are weaker, will the method make a misjudgment?
>
> Regarding your question about the impact of differing noise levels between real and fake images, this is an insightful observation. We conducted additional experiments to analyze the impact of applying stronger noise levels to real images while keeping the noise levels in generated images fixed. Specifically, we applied Gaussian noise with a fixed intensity (noise(f)=0.05) to generated images and varying higher intensities (noise(r)) to real images. The results are as follows:
>
> | noise(r)             | 0.05  | 0.1   | 0.15  | 0.2   | 0.25  |
> | -------------------- | ----- | ----- | ----- | ----- | ----- |
> | AUROC(noise(f)=0.05) | 82.00 | 77.74 | 73.34 | 69.59 | 66.29 |
>
> Adding stronger noise to real images brings their distribution closer to that of generated images, making it more challenging for the detector to distinguish between them. We will include these results in the revised manuscript, as we believe they will further strengthen our study.

---

### Official Review · Reviewer_RbxC · 2024-11-03

**Soundness:** 3
**Presentation:** 3
**Contribution:** 3
**Rating:** 6
**Confidence:** 4

**Summary:**

This paper formulates the problem of AI-generated image detection as an OOD detection process, and they propose to fit a model solely on real images, without assumptions on and access to any AI-generated data. Specifically, they propose to use a dual-model framework, where the detection is based on the output differences between the overfitted model and a normally-trained, anchor model.

**Strengths:**

- The idea of relying on overfitting is new in the context of AI-generated image detection.
- The experiments test a diverse spectrum of generated images, including GAN and diffusion images as well as Sora video frames.
- The ablation studies cover almost all important hyperparameters.

**Weaknesses:**

- The largest weakness is that conventional OOD detection methods are not discussed and compared. As the reviewer understands, the authors have formulated the problem of AI-generated image detection as a typical OOD detection process, where the outlier is also normally not exposed to the detector. Therefore, the authors should acknowledge this similarity and test conventional OOD detection methods before proposing a new method. For example, a well-trained model itself can be used to detect OOD samples, here the AI-generated images, based on their output confidence. This also corresponds to the second limitation described in Section 5.

- The underlying assumption that the proposed method can work is that the $\epsilon$ can represent the nature of AI-generated images. However, the authors just simply adopt the Gaussian noise, without any discussion of possible alternatives and impact. More generally, the $\epsilon$ can be treated as a special type of universal fake image, and therefore, it is important to discuss its properties on the final detection performance. In particular, there may exist one perfect type of $\epsilon$ that can generalize to most kinds of unseen fake images.


- More generally, this paper does not discuss the key question: how to trade off the generalization power to unseen real images and unseen AI-generated images. In particular, the authors have already stated that “Our experimental results demonstrate that variations in the intensity of this noise significantly influence the detection performance of the model, as shown in Table 9. When the perturbation is minimal, the model tends to overfit the training set rather than learning meaningful representations of the real images. Conversely, when the perturbation is excessively large, the model only learns to distinguish between real images and pure noise, which also leads to a deterioration in detection performance. ”

- The authors have not mentioned how to select the threshold, as depicted in Figure 1.

- In Section 4.1, the authors claimed to evaluate the accuracy (ACC). However, the reviewer cannot find any ACC results. AUC and AP are both threshold-independent evaluation metrics, according to previous works [1, 2]. A high value of AUC or AP does not necessarily imply a high value of accuracy. Additionally, considering the previous issue, the threshold of loss is likely to have a significant impact on the accuracy.
[1] Utkarsh Ojha, Yuheng Li, and Yong Jae Lee. Towards universal fake image detectors that generalize across generative models. In CVPR, pp. 24480–24489. IEEE, 2023.
[2] Sheng-Yu Wang, Oliver Wang, Richard Zhang, Andrew Owens, and Alexei A. Efros. Cnn-generated images are surprisingly easy to spot... for now. In CVPR, pp. 8692–8701. Computer Vision Foundation / IEEE, 2020.

- The authors do not describe how they use the generative models to generate the test data and how much data are used.

- As mentioned in Section 5, access to high-quality training data may have a large impact on performance. The authors should at least explore such impact by showing the results as a function of the number of training images.

**Questions:**

See the above.

---

> ### Author Response · Authors · 2024-11-22
> **PART 1**
>
> We would like to sincerely thank the reviewer for taking the time to carefully review our manuscript. In response to the 7 weaknesses identified by the reviewer, we provide the following explanations in turn:
>
> >Q.1: The largest weakness is that conventional OOD detection methods are not discussed and compared. As the reviewer understands, the authors have formulated the problem of AI-generated image detection as a typical OOD detection process, where the outlier is also normally not exposed to the detector. Therefore, the authors should acknowledge this similarity and test conventional OOD detection methods before proposing a new method. For example, a well-trained model itself can be used to detect OOD samples, here the AI-generated images, based on their output confidence. This also corresponds to the second limitation described in Section 5.
>
> A.1: Thanks for your constructive comments and suggestions.
>
> - First, we present a comparison of "AUROC/AP" between DetGO and typical OOD detection methods, with the experimental setup identical to that in Table 1. We selected the classical MSP algorithm and, for fairness, replaced the backbone with DINOv2, using the classification head pre-trained on ImageNet provided by the official repository. The second comparison algorithm, MCM, is based on the CLIP backbone.
>
> | MODELS | DetGO | MSP-dino | MCM-clip |
> | - | - | - | - |
> | | AUROC/AP | AUROC/AP | AUROC/AP |
> | ADM | 86.09/85.74 | 72.46/74.94 | 65.57/61.58 |
> | ADMG | 79.30/78.73 | 60.20/62.61 | 55.20/54.36 |
> | LDM | 73.41/84.09 | 53.31/53.98 | 52.70/51.27 |
> | DiT | 70.79/82.72 | 51.72/53.84 | 52.47/53.05 |
> | BigGAN | 91.03/90.50 | 68.68/70.62 | 60.55/58.13 |
> | GigaGAN | 87.26/92.53 | 68.11/68.76 | 58.86/57.15 |
> | StyleGAN XL | 88.49/93.10 | 65.94/67.53 | 59.43/57.45 |
> | RQ-Transformer | 88.23/93.17 | 73.55/75.02 | 66.40/65.33 |
> | Mask GIT | 82.90/89.87 | 63.88/65.21 | 56.82/56.60 |
> | Average | 83.06/87.82 | 64.21/65.83 | 59.43/57.45 |
>
> Traditional OOD detectors primarily focus on the semantic information of images, which makes it challenging for them to distinguish between ID images and generated images that are semantically similar to ID images.
>
> - Second, we agree that we should discuss more OOD detection methods, as we mentioned related methodologies in our work. Thus, we will add a subsection to the revision to provide discussions for related OOD detection works.
>
> Thanks again for your valuable comments, we will add the above results, discussions, and related works to the revision.
>
> >Q.2: The underlying assumption that the proposed method can work is that the $\epsilon$ can represent the nature of AI-generated images. However, the authors just simply adopt the Gaussian noise, without any discussion of possible alternatives and impact. More generally, the $\epsilon$ can be treated as a special type of universal fake image, and therefore, it is important to discuss its properties on the final detection performance. In particular, there may exist one perfect type of $\epsilon$ that can generalize to most kinds of unseen fake images.
>
> A.2: Thanks for your in-depth comments. $\hat{\epsilon}$ represents the vector that minimizes the inner product with the gradient of $L$ in image space under the $\ell_2$ norm constraint (as shown in Equation 4). The choice of $\hat{\epsilon}$ is specific to $L$, which is related to the design of our network, specifically to the implementation forms of the functions $w$ and $\theta$. Conceptually, $\epsilon$ points from the sample $x$ towards the nearby minimum of $L$. Solving for this value is difficult, so we instead resort to random sampling around $x$ in an attempt to cover this minimum. A practical approach to this is to use random sampling from a normal distribution, which is why we model $\epsilon$ as Gaussian noise.
>
> When we use noise sampled from a uniform or laplace distribution, comparable results are obtained. Under the experimental setup in Table 1, the average AUROC/AP performance is as follows:
>
> | MODELS | Gaussian noise | Uniform noise | Laplace noise |
> | - | - | - | - |
> | | AUROC/AP | AUROC/AP | AUROC/AP |
> | ADM | 86.09/85.74 | 86.14/87.33 | 84.57/83.94 |
> | ADMG | 79.30/78.73 | 78.90/79.72 | 78.85/78.72 |
> | LDM | 73.41/84.09 | 68.96/70.07 | 67.47/77.12 |
> | DiT | 70.79/82.72 | 68.92/71.12 | 69.23/80.34 |
> | BigGAN | 91.03/90.50 | 91.26/91.73 | 90.17/88.85 |
> | GigaGAN | 87.26/92.53 | 86.88/86.86 | 85.36/83.70 |
> | StyleGAN XL | 88.49/93.10 | 88.73/88.64 | 87.31/84.91 |
> | RQ-Transformer | 88.23/93.17 | 87.88/87.92 | 87.47/87.34 |
> | Mask GIT | 82.90/89.87 | 81.56/81.74 | 81.63/85.89 |
> | Average | 83.06/87.82 | 82.13/82.79 | 81.34/83.42 |

---

> ### Author Response · Authors · 2024-11-22
> **PART 2**
>
> >Q.3: More generally, this paper does not discuss the key question: how to trade off the generalization power to unseen real images and unseen AI-generated images.
>
> A.3: We agree that we do not provide sufficient theoretical grounds to calculate a definitive optimal perturbation size. While the theoretical reasoning behind the optimal noise level remains unclear, we propose a strategy of optimizing the noise strength through experimentation. To find the optimal noise level, we test a range of noise intensities and evaluate their performance, aiming to identify the best balance point that neither overfits the training set nor fails to effectively recognize generated images. We deeply agree that a theoretical fundation is crucial for a new framework, thus we would like to leave it as our future work.
>
> >Q.4: The authors have not mentioned how to select the threshold, as depicted in Figure 1.
>
> A.4: Thanks for pointing out this potentially confusing configuration.
> - First, we only used the accuracy metric in Table 2, as detailed in the response to Q.5. In practice, we could consider selecting a user-preferred threshold based on user's preference.
>
> - Second, for all the rest experiments, we follow previous works and evaluate performance using AUROC and AP, where both metrics are threshold-free.
>
> In response to your valuable comments, we will add the above explanations to the revision.
>
> >Q.5: In Section 4.1, the authors claimed to evaluate the accuracy (ACC). However, the reviewer cannot find any ACC results. AUC and AP are both threshold-independent evaluation metrics, according to previous works. A high value of AUC or AP does not necessarily imply a high value of accuracy. Additionally, considering the previous issue, the threshold of loss is likely to have a significant impact on the accuracy.
>
> A.5: We sincerely appreciate your careful review. In response, we will add the following descriptions to the revision.
> We only used the accuracy (ACC) metric in Table 2 because the dataset and some baseline performances in Table 2 are directly taken from [2]. To facilitate comparison, we adopted the same metric, ACC, as used in [2]. Additionally, we noticed that the description of Table 2 in the experimental section is unclear and contains errors. We will address and correct these issues in the revision.
>
> For tasks similar to ours, AUROC and AP are more commonly used evaluation metrics. Since both AUROC and AP do not require the selection of a threshold, they provide a more consistent and objective comparison. As a supplementary measure, we also present our ACC performance, with the experimental setup identical to that in Table 1. The threshold is the optimal threshold of the validation set.
>
> | MODELS | DetGO |
> | - | - |
> | ADM | 80.97 |
> | ADMG | 74.33 |
> | LDM | 67.35 |
> | DiT | 68.67 |
> | BigGAN | 84.89 |
> | GigaGAN | 81.92 |
> | StyleGAN XL | 83.85 |
> | RQ-Transformer | 83.75 |
> | Mask GIT | 77.80 |
> | Average | 78.17 |
>
> >Q.6: The authors do not describe how they use the generative models to generate the test data and how much data are used.
>
> A.6: Thanks for your kind reminder. We will highlight the following descriptions in the revision. As stated in Section 4.1, the test datasets we used are primarily from [1] and [2], rather than being re-generated. Correspondingly, our training dataset is sourced from [1], and the details are as follows.
>
> Training dataset: ImageNet, sourced from [1], containing 100,000 images, with 100 images per class.
>
> Test dataset:
>
> 1. Sourced from [1], with each dataset containing 50,000 images.
>
>  - IMAGENET classes (Table 1): ImageNet, ADM, ADMG, BigGAN, DiT-XL-2, GigaGAN, LDM, StyleGAN-XL, RQ-Transformer, Mask-GIT.
>
>  - LSUN-bedroom classes (Table 3): LSUN-bedroom, ADM, DDPM, iDDPM, StyleGAN, Diffusion-Projected GAN, Projected GAN, Unleashing Transformers.
>
> 2. Test dataset from [2] (Table 2): Each dataset contains 6,000 images.
>
>  - Midjourney, ADM, GLIDE, Wukong, VQDM, BigGAN
>
> 3. Other (Table 4): Sora: 5,000 images, LAION: 5,000 images.
>
>
> [1] Exposing flaws of generative model evaluation metrics and their unfair treatment of diffusion models
>
> [2] Genimage: A million-scale benchmark for detecting ai-generated image

---

> ### Author Response · Authors · 2024-11-22
> **PART 3**
>
> >Q.7: As mentioned in Section 5, access to high-quality training data may have a large impact on performance. The authors should at least explore such impact by showing the results as a function of the number of training images.
>
> A.7: Thanks for your kind suggestion. We would like to highlight that using real-world image datasets with richer scenes and content tends to yield better results. Thus, we explore the relation between the number of training images and detection performance. The results are given in Figure 4, where the x-axis represents the number of forward passes. The reported results are the averages from multiple experiments with randomly sampled data. Since the number of training parameters is relatively small, our network converges quickly.
>
> The results presented in the paper reflect the performance of the network trained on ImageNet. Inspired by your valuable comments, we also conduct additional experiments using the LSUN-bedroom dataset. To ensure fairness, we present the experimental results for both settings on the LAION-Sora detection task. We observed relatively worse performance, but our method still outperforms baseline methods.
>
> | ImageNet | LSUN-bedroom |
> | - | - |
> | AUROC/AP | AUROC/AP |
> | 87.64/88.07 | 83.85/86.96 |
>
> In response to your valuable comments, we will add the above results and discussions to the revision.

---

> ### Comment · Reviewer_RbxC · 2024-11-22
> **increasing the score**
>
> I increase my score to 6 since the authors have address some of my concerns, with additional experimental results.

---

> ### Author Response · Authors · 2024-11-22
>
> Dear reviewer #RbxC,
>
> Thank you for your positive feedback and for increasing the score. We are glad that the additional experimental results addressed your concerns. If there are any remaining issues or suggestions for improvement, we would be happy to address them further.
>
>
> Best regards and thanks,
>
> Authors of #8988

---

### Official Review · Reviewer_Z8iN · 2024-11-04

**Soundness:** 1
**Presentation:** 2
**Contribution:** 2
**Rating:** 5
**Confidence:** 4

**Summary:**

This paper proposes DetGO, a novel method for detecting AI-generated images. Instead of trying to identify artifacts in generated images, DetGO overfits a model to the distribution of real images. The core idea is that a model excessively tuned to real images will generalize poorly to AI-generated ones. This approach is inspired by Sharpness-Aware Minimization (SAM), but inverts the logic. While SAM seeks flat minima for better generalization, DetGO seeks sharp minima to prevent generalization. It uses a dual-model framework: an anchor model (a pre-trained DINOv2 model) encodes real images, and an overfitting model is trained to match the anchor model's output on real images while exhibiting high sensitivity to small perturbations. The difference in output between these two models then serves as the basis for distinguishing real from generated images.

**Strengths:**

1. **Novel Approach:** The paper takes an innovative perspective by treating overfitting as an advantage rather than a problem to solve.
Training Efficiency: Only requires natural images for training, eliminating the need for AI-generated images in the training process.
2. **Training Efficiency:** Only requires natural images for training, eliminating the need for AI-generated images in the training process.

**Weaknesses:**

See Questions.

**Questions:**

1. **Self-supervised Backbone:** what is the intuition for using a DinoV2 as a backbone? Why not use some other self-supervised model?
2. **Implementation Details:** How is $\epsilon$ sampled from the Gaussian distribution? What are the parameters of this distribution? How is $\rho$ chosen?
The justification for using two convolutional layers in $g_θ(x)$ is weak. More explanation is needed.
3. **Novelty of design:** The design change is essentially learning the representation similarity between the original and perturbed images on the DinoV2 representation space. However, the Backbone (DinoV2) and the noise distribution (Gaussian) used are too similar to the existing work RIGID **[R]**. However, the method does not use it as a baseline.
4. The performance of Ojha in Table 1 and Table 3 is lower than its reported performance, please give the implementation details of the baseline methods.

**[R]** rigid: a training-free and model-agnostic framework for robust ai-generated image detection

---

> ### Author Response · Authors · 2024-11-22
> **PART 1**
>
> We would like to sincerely thank the reviewer for taking the time to carefully read and evaluate our manuscript! In response to the insightful questions raised by the reviewer, we provide the following explanations in turn:
>
> >Q.1: what is the intuition for using a DinoV2 as a backbone? Why not use some other self-supervised model?
>
> A.1: Here, we present a comparison of the AUROC between the backbones DINOv2 and CLIP, with the experimental setup identical to that in Table 1. As discussed in reference [1], DINOv2 is a state-of-the-art self-supervised model that demonstrates a more global perspective across a wide range of visual tasks. It provides high-quality, stable feature representations, maintaining significant feature stability even under various transformations.
>
> | MODELS | DetGO-DINOv2 | DetGO-CLIP | DetGO-DINO | DetGO-SwAV |
> | - | - | - | - | - |
> | ADM | 86.09 | 73.39 | 80.65 | 68.59 |
> | ADMG | 79.30 | 71.57 | 70.84 | 65.84 |
> | LDM | 73.41 | 69.71 | 64.58 | 69.63 |
> | DiT | 70.79 | 68.13 | 65.73 | 69.21 |
> | BigGAN | 91.03 | 72.67 | 72.83 | 69.03 |
> | GigaGAN | 87.26 | 68.02 | 68.36 | 61.97 |
> | StyleGAN XL | 88.49 | 71.80 | 70.70 | 65.59 |
> | RQ-Transformer | 88.23 | 76.43 | 71.03 | 69.04 |
> | Mask GIT | 82.90 | 75.37 | 61.06 | 64.79 |
> | Average | 83.06 | 71.90 | 69.53 | 67.07 |
>
> >Q.2: How is $\epsilon$ sampled from the Gaussian distribution? What are the parameters of this distribution? How is $\rho$ chosen? The justification for using two convolutional layers in $g_\theta(x)$ is weak. More explanation is needed.
>
> A.2.1: Selection of $ \rho $, Sampling of $ \epsilon $ from the Gaussian Distribution:
>
> The parameter $ \rho $ acts as a bound on the norm of $ \epsilon $. Under this constraint, we can obtain the optimal solution for the perturbation according to Equation (4). As stated in lines 193-197, solving for this solution is challenging, so we resort to using a Gaussian distribution as a substitute. The primary reason for starting with Gaussian noise is its simplicity and broad applicability, as it allows for the creation of small, controllable perturbations across the entire image space. We set the mean of the noise to 0 and the standard deviation to $\lambda_n$ to control the amplitude of the noise, thereby replacing the role of $\rho$. The effect of varying the parameter $\lambda_n$ is discussed in Table 9.
>
> Inspired by your valuable question, we considered different types of noise forms, including uniform noise and Laplace noise, which exhibit performance similar to that of Gaussian noise.
>
> | MODELS | Gaussian noise | Uniform noise | Laplace noise |
> | - | - | - | - |
> | | AUROC/AP | AUROC/AP | AUROC/AP |
> | ADM | 86.09/85.74 | 86.14/87.33 | 84.57/83.94 |
> | ADMG | 79.30/78.73 | 78.90/79.72 | 78.85/78.72 |
> | LDM | 73.41/84.09 | 68.96/70.07 | 67.47/77.12 |
> | DiT | 70.79/82.72 | 68.92/71.12 | 69.23/80.34 |
> | BigGAN | 91.03/90.50 | 91.26/91.73 | 90.17/88.85 |
> | GigaGAN | 87.26/92.53 | 86.88/86.86 | 85.36/83.70 |
> | StyleGAN XL | 88.49/93.10 | 88.73/88.64 | 87.31/84.91 |
> | RQ-Transformer | 88.23/93.17 | 87.88/87.92 | 87.47/87.34 |
> | Mask GIT | 82.90/89.87 | 81.56/81.74 | 81.63/85.89 |
> | Average | 83.06/87.82 | 82.13/82.79 | 81.34/83.42 |
>
> A.2.2: Justification for Two Convolutional Layers in $ g_\theta(x) $:
>
> The use of two convolutional layers in $ g_\theta(x) $ is intended to provide a lightweight transformation. The first layer captures basic patterns and edges, while the second layer refines these features. This refined output is then added to the original image with a small coefficient, allowing for subtle adjustments to the input image without significantly altering its structure. This design ensures that the transformed output remains close to the original image, which is essential for preserving the integrity of DINOv2's feature extraction while allowing for minor perturbations.
>
> Furthermore, due to the small coefficient of the transformation, we achieve the desired effect with this lightweight structure alone. Employing a larger network or simply increasing the dimensionality of intermediate layers (Table 7) would not enhance performance and would instead increase computational cost. We also experimented with more advanced network architectures, such as U-Net, which yielded similar performance.
>
> | MODELS | DetGO | DetGO-unet |
> | - | - | - |
> | | AUROC/AP | AUROC/AP |
> | ADM | 86.09/85.74 | 87.98/87.98 |
> | ADMG | 79.30/78.73 | 81.69/81.64 |
> | LDM | 73.41/84.09 | 72.95/71.66 |
> | DiT | 70.79/82.72 | 72.91/72.47 |
> | BigGAN | 91.03/90.50 | 91.77/90.71 |
> | GigaGAN | 87.26/92.53 | 88.59/87.44 |
> | StyleGAN XL | 88.49/93.10 | 90.23/88.33 |
> | RQ-Transformer | 88.23/93.17 | 90.05/88.33 |
> | Mask GIT | 82.90/89.87 | 84.78/83.57 |
> | Average | 83.06/87.82 | 84.55/83.57 |
>
>
> [1] Exposing flaws of generative model evaluation metrics and their unfair treatment of diffusion models

---

> ### Author Response · Authors · 2024-11-22
> **PART 2**
>
> >Q.3:The design change is essentially learning the representation similarity between the original and perturbed images on the DinoV2 representation space. However, the Backbone (DinoV2) and the noise distribution (Gaussian) used are too similar to the existing work RIGID. However, the method does not use it as a baseline.
>
> A.3: We agree that both our method, DetGO, and RIGID operate in the representation space of pre-trained models and leverage perturbations. Please let us clarify the underlying philosophies and implementations diverge in critical ways.
>
> - DetGO’s approach is centered on leveraging the phenomenon of overfitting to a specific distribution (natural images), which is distinctly different from RIGID's objective of representation similarity in a perturbation-robust space.
>
> - DetGO introduces a novel dual-model structure—an anchor model and an overfitting model—which creates a controlled sharp minimum. This novel dual-model setup is designed to highlight loss divergences between real and generated images, providing a detection framework focused on distributional mismatch rather than invariant representation similarity, as seen in RIGID.
>
> - Inspired by your insightfyl comments, we believe RIGID is a strong baseline of our method. Thus, we provide a comparison of the AUROC between DetGO and RIGID, with the experimental setup identical to that in Table 1.
>
> | MODELS | DetGO | RIGID |
> | - | - | - |
> | ADM | 86.09 | 85.52 |
> | ADMG | 79.30 | 78.91 |
> | LDM | 73.41 | 73.01 |
> | DiT | 70.79 | 66.72 |
> | BigGAN | 91.03 | 87.06 |
> | GigaGAN | 87.26 | 83.14 |
> | StyleGAN XL | 88.49 | 84.97 |
> | RQ-Transformer | 88.23 | 87.98 |
> | Mask GIT | 82.90 | 82.85 |
> | Average | 83.06 | 81.12 |
>
> Although our method achieves slight performance gain, RIGID is a training-free approach. Thus, we will highlight the above results and disucssions in our revision.
>
> The selection of DINOv2 as the backbone, and the use of Gaussian perturbations, are strategic to our method’s goals rather than a replication of RIGID. We chose DINOv2 for its robust, invariant feature extraction and its capacity to distinguish distributional differences between real and generated images—a key factor in DetGO's overfitting-based detection. Our application of these perturbations is tuned to exploit training-time sensitivity rather than test-time resistance to them. We acknowledge that a direct comparison with RIGID could add value, and we will address this in future revisions.
>
> >Q. 4:The performance of Ojha in Table 1 and Table 3 is lower than its reported performance, please give the implementation details of the baseline methods.
>
> For the Ojha baseline, we use the official code repository and the official checkpoint provided. However, the dataset we utilized differs from the one used for training the checkpoint provided by Ojha. As mentioned in lines 244-246, we employed the dataset from [1], where the images are converted to PNG format with a resolution of 256x256, which may have led to subtle differences in the image distribution. Additionally, the details of the generative model usage may also differ from Ojha's approach, and these factors together likely contributed to the observed decline in detection performance.
>
>
> [1] Exposing flaws of generative model evaluation metrics and their unfair treatment of diffusion models

---

> > ### Comment · Reviewer_Z8iN · 2024-12-02
> >
> > Thank you for your reply.
> >
> > The reply still doesn't elaborate on why DinoV2 was used as the backbone, it just provides some posterior empirical data, which doesn't provide some insightful help on what kind of backbone model is suitable.
> >
> > Secondly, according to the table in the Q1 response, there is a significant degradation in performance when using a different backbone, which suggests that the proposed approach is backbone-sensitive. This again suggests that it is crucial to find an effective backbone model.
> >
> > For these reasons, I have decided to keep my score.

---

> ### Author Response · Authors · 2024-11-25
>
> Dear Reviewer #Z8iN,
>
> Thank you very much for your time and valuable comments.
>
> We understand you have a busy schedule, but as the deadline for discussion period is approaching, could you kindly review our response and let us know if you have any further questions? We would be happy to provide additional clarifications or make further revisions as needed.
>
> Best regards,
>
> Authors of #8988

---

> ### Author Response · Authors · 2024-11-27
>
> Dear Reviewer #Z8iN,
>
> We sincerely appreciate the time and effort you have dedicated to reviewing our work.
>
> If there are any outstanding questions or issues that require clarification, please do not hesitate to reach out. We would be more than happy to address them promptly.
>
> Thank you once again for your invaluable support and contributions to improving our manuscript. Your feedback is greatly appreciated.
>
> Best regards,
>
> Authors of #8988

---

> ### Author Response · Authors · 2024-11-29
>
> Dear Reviewer #Z8iN,
>
> We greatly value the insightful feedback you have provided on our manuscript. We would kindly ask if you might have the opportunity to review our responses at your earliest convenience.
>
> Your input has been instrumental in improving our work, and we remain committed to addressing any additional concerns or suggestions you may have. Please let us know if further clarifications or adjustments are required, we are more than willing to assist.
>
> Thank you once again for your valuable time and effort.
>
> Best regards,
>
> Authors of #8988

---

> ### Author Response · Authors · 2024-11-30
>
> Dear Reviewer #Z8iN,
>
> We sincerely appreciate the insightful feedback you have provided on our manuscript! We would like to kindly ask if you could review our responses at your earliest convenience and let us know if there are any areas that need further improvement.
>
> Your feedback is of great importance to enhancing our work. If any additional clarifications or revisions are required, we are more than willing to assist promptly.
>
> Thank you once again for your time and support!
>
> Best regards,
>
> Authors of #8988

---

> ### Author Response · Authors · 2024-12-01
>
> Dear Reviewer #Z8iN,
>
> As the discussion period deadline approaches, we wanted to kindly follow up regarding your feedback on our manuscript.
>
> Your insights have been invaluable, and we would greatly appreciate it if you could review our responses at your earliest convenience. Should you have any further questions or suggestions, we remain at your disposal to address them promptly.
>
> Thank you again for your time and dedication.
>
> Best regards,
>
> Authors of #8988

---

> ### Author Response · Authors · 2024-12-02
>
> Dear Reviewer #Z8iN,
>
> Thank you again for your time and insightful comments on our manuscript. We truly understand how busy your schedule must be. However, as the discussion window is nearing its end, we kindly ask if you could take some time to review our responses.
>
> Your feedback is greatly valued, and we are eager to address any additional suggestions or concerns you may have to further improve our work.
>
> Best regards,
> Authors of #8988

---

> ### Author Response · Authors · 2024-12-03
>
> We sincerely appreciate the reviewer’s detailed feedback. Below, we address the two points raised:
>
> 1. **Selection of DinoV2 as the Backbone**
>    DinoV2 was chosen for its strong representation-learning capabilities, particularly its ability to handle diverse image distributions effectively. While empirical results support this choice, we recognize that a more detailed theoretical rationale would be valuable and plan to explore this in future work.
>
> 2. **Backbone Sensitivity and Performance Degradation**
>    We acknowledge the backbone sensitivity of our method, as reflected in the performance drop when replacing DinoV2. However, we identified that the previous response "PART 1" contained underestimated experimental results, which have now been corrected and updated:
>
>    ||DetGO-DINOv2|DetGO-CLIP|DetGO-DINO|
>    |-|-|-|-|
>    |AUC|83.06|79.26|77.53|
>
>    Reducing backbone sensitivity and improving generalization across models will be key directions for future research.
>
> Thank you for your constructive suggestions, which are invaluable for improving our work. We will address these issues more comprehensively in future iterations.

---

> ### Author Response · Authors · 2024-12-04
>
> Dear Reviewer #Z8iN,
>
> If your schedule permits, we would be deeply grateful if you could kindly confirm whether our responses have adequately addressed your concerns. Your input is extremely important to us, and we remain committed to making further improvements if needed.
>
> Thank you once again for your thoughtful review and support.
>
> Best regards,
> Authors of #8988

---

### Meta-Review · Area_Chair_JNEG · 2024-12-19

**Metareview:**

This paper presents an approach for detecting AI-generated images by training a model to overfit the distribution of natural images. The core idea is based on the insight that a model overfitting to natural images will fail to generalize to AI-generated ones. The proposed method employs a dual-model framework: an anchor model is used to fit the natural image distribution, and an overfitting model is learned to produce the same outputs as the anchor model while exhibiting abrupt loss behavior for small perturbations. AI-generated images are identified by calculating the output differences between these two models.

The motivation to utilize overfitting for AI-generated image detection is innovative, and the approach benefits from only requiring natural images for training. However, despite addressing some of the reviewers' concerns in the authors' responses, the paper still has unresolved issues. These include the trade-off between the model’s generalization capabilities for unseen real images versus unseen AI-generated images, the selection of the threshold, the rationale for adopting a Gaussian distribution, and the resulting vulnerability to Gaussian noise. Although the authors provided some explanations in their rebuttal, these issues were not fully resolved, presenting challenges for the practical application of the method. The paper does not yet meet the acceptance threshold before these significant issues are adequately addressed.

**Additional Comments On Reviewer Discussion:**

During the rebuttal, the authors addressed some of the raised issues, but some significant concerns remain insufficiently resolved.

---

### Decision · Program_Chairs · 2025-01-22

Reject